# Subcutaneous Granuloma Annulare vs. Subcutaneous Vascular Malformations in Children: A Diagnostic Challenge

**DOI:** 10.3390/children10020362

**Published:** 2023-02-11

**Authors:** Besiana P. Beqo, Paolo Gasparella, Christina Flucher, Sebastian Tschauner, Iva Brcic, Emir Q. Haxhija

**Affiliations:** 1Department of Paediatric and Adolescent Surgery, Medical University of Graz, A-8036 Graz, Austria; 2Boston Children’s Hospital and Harvard Medical School, Boston, MA 02115, USA; 3VASCERN VASCA European Reference Centre, Bichat-Claude Bernard Hospital, 75018 Paris, France; 4Division of Paediatric Radiology, Department of Radiology, Medical University of Graz, A-8036 Graz, Austria; 5Institute of Pathology, Medical University of Graz, A-8036 Graz, Austria

**Keywords:** granuloma annulare, subcutaneous granuloma annulare, self-limiting, children, low-flow subcutaneous vascular malformations, venous malformations, lymphatic malformations

## Abstract

Objectives. There are various subcutaneous lesions in children and often there is difficulty in obtaining an accurate diagnosis by non-invasive diagnostic procedures. Subcutaneous granuloma annulare (SGA) is a rare granulomatous disease that, even after imaging, is often mistaken for a low-flow subcutaneous vascular malformation (SVM). This study aimed to accurately identify clinical and imaging clues to distinguish SGA from low-flow SVM. Methods. We retrospectively analyzed complete hospital records of all children with a confirmed diagnosis of SGA and low-flow SVM who underwent MR imaging at our institution from January 2001 to December 2020. Their disease history, clinical and imaging findings, management, and outcome were evaluated. Results. Among 57 patients with granuloma annulare, we identified 12 patients (nine girls) with a confirmed SGA diagnosis who underwent a preoperative MRI. Their median age was 3.25 years (range 2–5 years). Of 455 patients diagnosed with vascular malformations, 90 had malformations limited to the subcutaneous area. Among them only 47 patients with low-flow SVM were included in the study and further analyzed. Our SGA cohort had a female predilection (75%) and a short history of lump appearance of 1.5 months. SGA lesions were immobile and firm. Before MRI, patients underwent initial evaluation by ultrasound (100%) and X-ray (50%). Surgical tissue sampling was performed in all SGA patients to establish a diagnosis. All 47 patients with low-flow SVM were diagnosed correctly by MRI. A total of 45 patients (96%) underwent surgical resection of the SVM. A careful retrospective review of imaging findings of patients with SGA and SVM showed that SGA present as homogenous lesions in the shape of an epifascial cap with a typical broad fascial base extending towards the subdermal tissue in the middle of the lesion. In contrast, SVMs always present with variable-sized multicystic or tubular areas. Conclusions. Our study shows clear clinical and imaging differences between low-flow SVMs and SGA. SGA presents characteristically in the shape of a homogenous “epifascial cap,” which distinguishes these lesions from multicystic heterogenous SVMs.

## 1. Introduction

Subcutaneous granuloma annulare (SGA) is a benign, subcutaneous disease of an unknown etiology that exclusively affects young children [1]. This uncommon disease is characterized by the sudden appearance of firm, immobile, solitary, or multiple subcutaneous lumps usually located over the bony prominences of the extremities or the scalp [2]. These lesions are painless and seem to spontaneously self-resolve in about two years without requiring any type of treatment [3,4].

Reaching an informed and accurate diagnosis in children with SGA commonly presents a real clinical challenge for a medical examiner [5]. The challenge becomes evident when one tries to develop a reasonable differential diagnosis for the patient’s symptoms. Considering all the possible diverse pathologies and their overlapping clinical behaviors, SGA is most commonly mistaken for subcutaneous low-flow vascular malformations (SVM) [6,7].

Vascular malformations are not neoplasms but inborn errors of angiogenesis that grow proportionally with the child and usually persist lifelong [8,9]. They can be clinically depicted as venous, capillary, lymphatic, arteriovenous, or mixed [10]. In addition, vascular malformations can be classified as low-flow or high-flow, depending on their hemodynamic characteristics. The low-flow vascular malformations are composed of nonarterial components and can either be capillary, venous, lymphatic, or any combination of these components [11]. Depending on their variable clinical appearance and presentation, subcutaneous low-flow vascular malformations can also substantially vary in imaging appearance.

To date, little to nothing has been reported on clinical imaging characteristics that could distinguish SGA from SVM. This knowledge gap gives rise to diagnostic confusion, and reaching a definitive diagnosis in these cases often requires further invasive diagnostic procedures [5,12,13,14]. As a consequence, regardless of the harmless and self-limiting nature of the SGA, accurate diagnosis is most commonly achieved only by histopathologic analysis after an incisional or excisional biopsy of the child’s lump(s) [15,16,17].

This study aims to analyze the distinctive clinical and imaging characteristics of SGA and SVM and highlight a rational diagnostic-pathway approach to patients suspected of having SGA.

## 2. Patients and Methods

This is a cross-sectional, explorative, retrospective study of patients 0–18 years old diagnosed with SGA or low-flow SVM who underwent MR imaging to evaluate their subcutaneous lesions between 1 January 2001, and 31 December 2020. All patients with a confirmed diagnosis of SGA and low-flow SVM were eligible to be included in our study. A positive histopathologic analysis report for SGA was accepted as a confirmed SGA case.

The specific vascular malformation category for the SVMs was assigned based on the evaluation of an expert clinical examination, medical history, lab, and imaging diagnostics. Using the classification of the International Society for the Study of Vascular Anomalies (ISSVA), SVM patients were grouped into arterio-venous malformations (AVM), venous malformations (VM), lymphatic malformations (LM), and mixed malformations [10].

Exclusion from the study followed all patients clinically diagnosed by pathognomonic clinical signs, such as lesion pulsation for AVMs or bluish compressible lesions for VMs, and patients who did not undergo an MRI evaluation of their lesion(s).

The electronic hospital records (medocs) of all patients included in the study were retrospectively reviewed regarding the patient’s disease history, management, and outcome. The information identified and analyzed for each case included: sex, age at the initial encounter, the location of the lesion(s), the number of the lesion(s), associated pain, the clinical description of the lesion(s), the report of trauma prior to the appearance of the lesion(s), the time between the first presentation at our department and treatment starting, the diagnostic imaging and lab work performed, the histopathologic report provided, the type of treatment, the recurrence after treatment, and the time to follow-up. Descriptive statistical analysis was performed. This study has been approved by the Ethics Committee of the Medical University of Graz (Approval code: EK-No. 33-126 ex 20/21; date: 20 January 2021).

## 3. Results

A total of 57 patients presented with granuloma annulare at our department during the study period. Twenty-eight of these patients had a confirmed diagnosis of SGA. However, only 12 of them had a preoperative evaluation by magnetic resonance imaging (MRI) and, therefore, qualified to be included in this study.

Among 455 patients diagnosed with various vascular malformations at our department during the study period, 90 patients presented with a vascular malformation located just beneath the skin, in the subcutaneous tissue. Twenty-three of them were diagnosed clinically and therefore were excluded from further analysis; nine presented with pulsatile subcutaneous lesions, making the clinical diagnosis of an arteriovenous malformation obvious (Figure 1); and 14 had a history of bluish compressible subcutaneous lesions that clinically suggested the diagnosis of a venous malformation (Figure 2).

Out of 67 patients with a confirmed diagnosis of an SVM, only 47 had MRI records and were further qualified to be included in this study. 

The demographic characteristics of the patients included in this study are presented in Table 1. Table 2 shows the location of the lesions in patients with SGA and low-flow SVM that underwent MR imaging.

### 3.1. Patients with Subcutaneous Granuloma Annulare

#### 3.1.1. Clinical Presentation

A strong female predilection of 3:1 is noted in our SGA cohort. All patients with SGA who received an MRI were preschool children from two to five years old, with a median age of 3.25 years at their first presentation. Seven SGA patients presented with singular lesions, three patients had two lesions each, and two patients had three lesions each. All our patients with SGA on the scalp had multiple lesions. The pretibial area was noted to be the most commonly presented location for SGA, as almost half of our patients had at least one pretibial lesion. All SGA lesions were clinically described as firm, nontender, and immobile subcutaneous lumps without any signs of inflammation. No overlying cutaneous abnormalities were reported, and all the children were described as otherwise healthy. The SGA were said to have appeared on an average of one and a half months (range 0.5–3 months) before the time of encounter, and previous trauma at the site of SGA was reported in five cases (42%).

#### 3.1.2. Presentation on X-ray

An X-ray examination was performed in half of our SGA patients (i.e., 6/12 or 50%) because of the firm and immobile nature of the lumps and the trauma history reported by parents. While excluding any types of bony abnormalities, a homogenous subcutaneous soft tissue shadow was regularly reported (Figure 3).

#### 3.1.3. Presentation on Ultrasound

All SGA patients received an ultrasound examination (US) at their first encounter. On the US, SGA were described as nonspecific mass(es) with interacting hypo- and hyper-echoic zones, ill-defined borders, and mild hypervascularization (Figure 4).

#### 3.1.4. Presentation on MRI

Because the etiology of the subcutaneous lesions remained unclear, all patients were referred for further evaluation by MRI. However, the reported results of all 12 MRI examinations neither excluded malignancy nor determined the accurate diagnosis. On the MRI, the SGA was consistently described as limited to the subcutaneous tissue without invasion of the muscle’s fascia. SGA was reported to be isointense relative to the muscles on T1-weighted images and hyperintense relative to the muscles on T2-weighted images. The lesions enhanced after contrast agent administration. The primary differential diagnosis in all patients was low-flow vascular malformation, followed by fibromatosis, fasciitis, panniculitis, and post-traumatic fat necrosis. SGA was considered as a differential diagnosis in one case only.

#### 3.1.5. Invasive Diagnostic Procedures

Since a possible malignant nature of the lesion(s) could not be excluded, all children were referred for invasive tissue sampling. The preoperative lab work included a complete blood count and a basic metabolic panel, which resulted uneventfully in every case. Seven children underwent an excisional biopsy, and five had an incisional biopsy performed; all resulted in the histopathologic diagnosis of SGA. The median follow-up time after the surgical intervention was 11 months (range 6–17 months), and no further interventions were needed.

### 3.2. Patients with Low-Flow Subcutaneous Vascular Malformations

#### 3.2.1. Clinical Presentation

In contrast to SGA, patients with SVM were mainly boys (60%) who presented at a median age of 5 years with a wide age range from birth to 18 years. The subcutaneous lesions in patients diagnosed with SVM were often clinically described as indolent, soft, and mobile. The lesions were described as bulging and tender in cases with sudden enlargement of the SVM after intracystic bleeding. 

While SGA patients regularly presented with a firm, nontender lump persisting for at least one month, patients with SVM presented with various medical histories and clinical presentations. Twenty-two patients with SVM (47%) presented at their first encounter between birth and the third year of life. Only four (18%) of these patients had a sudden enlargement of the affected area of the body, which was caused by intracystic bleeding (Figure 5). Parents were aware of their child’s vascular malformation from birth or the first month of their life in 15 cases (68%). Eight patients (17%) presented between their fourth and 10th year of life. Among them, five (63%) patients had a sudden enlargement of the lesions with intracystic bleeding.

Seventeen patients (36%) presented after their 10th year of life, and nine (53%) of these patients presented after a sudden enlargement of their lesions. Altogether, 18/47 patients (38%) with only subcutaneous extension of their low-flow vascular malformations presented after a sudden enlargement of their SVM due to intracystic bleeding, most of which occurred in patients older than three years. The most common location for the SVM was the upper extremity; in particular, the axillary and upper arm region (Figure 6), where more than half of the SVM of our cohort were located, as shown in Table 2.

#### 3.2.2. Presentation on Ultrasound

At their initial encounter, each SVM patient underwent an ultrasound examination. The SVMs were mainly described as ill-defined masses of heterogeneous echotexture with visible multiple cystic spaces within. In cases that presented with sudden enlargement and recent or past bleeding into the cysts, a clear-cut sedimentation line was noted through the cyst representing the clot retraction as a pathognomonic sign of intracystic bleeding (see Figure 5). In five patients with SVM, no clear cystic areas were described during the ultrasound examination.

#### 3.2.3. Presentation on MRI

In all 47 cases, the MRI showed that the lesions consisted of multicystic or tubular vascular areas, and the diagnosis of low-flow SVM was considered in all cases. Figure 7 and Figure 8 illustrate the findings in one patient with a confirmed LM who presented with a lesion located over the pretibial area clinically mimicking SGA.

#### 3.2.4. Invasive Diagnostic Procedures

All except two patients with low-flow SVM underwent a complete surgical excision. In three of these patients, a recurrence of the SVMs was observed, and a second surgical intervention was needed for the complete removal of the SVM. The median follow-up time after surgical intervention in patients with SVMs was three years (range 2–5 years).

### 3.3. Retrospective Image Analysis

A retrospective review of all MR images showed that all MRIs of patients with low-flow SVMs showed multicystic and/or tubular lesion appearance, whereas MRIs of patients with SGA never showed any cyst-shaped structures. An SGA presents as a raised-rounded homogenous mass that projects over the surface of the muscle’s fascia without invading the underlying tissue (Figure 9). These lesions have a typical broad circular base laying on the fascia and a raised, continuous, irregular curved surface that extends from the deep fascia towards the more superficial tissues. One can easily describe the shape of an SGA as an island rising from the ocean. We recently named this characteristic shape “the Epifascial Cap,” the hallmark shape of an SGA (Figure 10) [18]. The enhancement of contrast material seen in these lesions during the MRI is homogenous and should not be mistaken for a low-flow SVM (Figure 11). 

Based on the patients’ medical history, clinical investigations, laboratory, and imaging findings, we have developed an algorithm of care for children with subcutaneous lesions suspected to be SGA, showed in Figure 12.

## 4. Discussion

Up to date, SGA is not easy to diagnose [2,5,19]. The sudden appearance, persistence, and growth of one or more subcutaneous lumps in a child usually incites parents to seek help from a physician [20,21,22]. These patients often see a number of specialists, such as pediatricians, pediatric surgeons, pediatric orthopedic surgeons, or pediatric dermatologists [23]. Most of the time, thorough diagnostic work-up is completed in order to identify the entity of an unclear subcutaneous lesion. Unfortunately, due to the present ambiguity, surgical intervention is frequently performed in patients with SGA which would not require any intervention [23,24,25]. 

Based on our study’s results, we propose a management algorithm for SGA in children. To provide appropriate recommendations for further treatment, clinicians must be knowledgeable about the clinical characteristics of the SGA and low-flow SVM, such as the exact history of the lesions, their softness, firmness, mobility, pain, and their predilected anatomic areas, as well as their imaging characteristics [21,26]. 

The first step towards an accurate diagnosis is taking a comprehensive medical history and a detailed physical examination, which are critical in evaluating a pediatric patient with a painless subcutaneous lump. SGA is usually located on trauma-exposed bony prominences of the lower extremities, scalp, and the ulnar side of the forearms. They often have an unclear history, with a relatively sudden appearance a few weeks to a couple of months ago. The lumps are firm, non-tender, and non-mobile. They present with no overlying skin abnormalities and show no signs of inflammation. Our 20-year cohort supports these findings and that the SGA is usually found in preschool-age children and most commonly in girls, confirming the female predisposition of this disease [14,15,23,27].

Low-flow SVMs are often present at birth but can become apparent also in the first or second decade of life. They grow in proportion with the child’s growth until puberty and can expand in response to certain stimuli such as trauma, infection, or pregnancy [11,28,29,30]. Venous malformations (VMs) are the most common type of vascular malformations, and their clinical presentation is variable. They may appear as a group of ectatic and dysplastic superficial veins or, more frequently, as deeper, real masses in the soft tissue with a soft, bluish, compressible appearance on the superficial skin [31,32]. However, the consistency of these lesions can increase due to the formation of internal clots, and they can appear as firm and immobile lumps, making it challenging to distinguish clinically from an SGA. They are most commonly found on the face, upper extremity, or trunk. LMs are often evident at birth and their most usual locations are the neck, the axillary region, and the mediastinum. LMs are slow-growing lesions that can enlarge significantly, leading to distortion of anatomy, especially of the soft tissues and bones of the face and trunk [28,31,32,33,34,35]. Most cases of SVMs are clinically obvious, and because of the specific time of their appearance and/or anatomic predilected areas, they do not pose diagnostic difficulties when differentiating from a rare SGA. However, when a diagnosis is not clear, we recommend referring the pediatric patient for routine complete blood work and US of the lesion, as the vast majority of subcutaneous lesions in children can be accurately evaluated with sonographic imaging [36,37]. Imaging evaluation allows us to provide appropriate recommendations concerning the prospect of the subcutaneous lesion(s). Should we observe them, we have to decide if the lesion requires observation and follow up, if the lesion needs excision but there is no urgency, or if there is a need for a prompt intervention to clarify the entity of the lesion(s).

On ultrasound, an SGA appears as an epifascial subcutaneous soft tissue mass, hypoechogenic in the center, with a hyperechoic zone in the periphery and mild vascularization. This lesion projects over the surface of the muscle’s fascia without invading it. Supposing that during the ultrasound evaluation, it is possible to recognize the epifascial cape shape of the lesion, we recommend using SGA as your primary working diagnosis and referring the child for a clinical follow-up with a repeat US of the lump in 4 weeks (Figure 12).

On ultrasound, VMs appear as well margin masses with a spongiform heterogenous echostructure which is hypoechoic compared to the surrounding tissues. Hypoechoic venous spaces and hyperechoic septa form the heterogenous echostructure of a VM. Sometimes on the US, it is possible to identify anechoic tubular structures that are recognized as vascular channels [32,38,39]. A pathognomonic sign that definitely aids in the diagnosis of VM is the presence of intralesional calcifications. Unfortunately, this sign is not frequent and its occurrence is reported variously from 9% to 16% in previously published series [32,38,39]. LMs appear as lesions that contain scattered, cystic formations of variable dimensions, with liquid content, separated by septa. LM are usually deformable, and the compression with the probe alters the shape of the cysts that never collapse entirely. The cystic spaces can be anechoic or have a variable degree of echogeneity due to intracystic bleeding or infections [32,40,41]. Given the extreme variety of ultrasound presentations of low-flow SVM and the difficulty of obtaining accurate information about the extent of the SVM, further imaging by MR is recommended. MRI is also recommended in case of doubt or if, upon returning for a clinical follow-up, it is noted that the size, consistency, and mobility of the lesion may have changed, or the family is now reporting other symptoms and signs associated with the lesion. 

An SGA is isointense relative to the muscles on T1-weighted images and hyperintense relative to the muscles on T2-weighted images. In addition, SGA is homogeneous in both T1- and T2-weighted MRI sequences and shows variable enhancement after contrast material injection. If the epifascial cap shape is visualized on MRI evaluation, consider SGA as the final working diagnosis, and request a clinical follow-up of the lump in 3 months with a repeat ultrasound evaluation to assess the size of the lesion.

VM on MRI will appear as multiple serpentine, tubular structures or amorphous dilated channels containing intermediate signal on T1-weighted images, high signal on T2-weighted images, intermediate signal on gradient echo sequences, and delayed enhancement on dynamic contrast-enhanced MRI [29,38,42,43,44,45,46,47]. LM appear as micro- or macrocystic spaces (depending on the type of the LM) that may contain fluid levels due to intracystic bleeding. Cysts will often be hyperintense on T2-weighted images and show no contrast enhancement [29,43,44,45,46,47].

If “worrying features” are noted during the follow-up visit, such as a recent increase in the size of the lump, deeper location relative to the fascia, and/or invasive growth patterns, then a prompt verification of the histopathologic diagnosis by biopsy is mandatory. Usually, low-flow SVM will require treatment at some point in life, while SGA spontaneously self-resolves in approximately two years’ time and does not need any treatment or surgical excision.

A limitation of our single-center study is the limited number of SGA cases identified during the 20-year study period. This is firmly attributed to the fact that SGA is a rare disease with an unknown etiology and is commonly overlooked by clinicians. Nevertheless, we hope our study results will give rise to other future investigations that can shed light on etiology of SGA and its current occurrence in the general population.

## 5. Conclusions

To the best of our knowledge, this is the first study that directly compared the clinical and imaging characteristics of rare SGA with low-flow SVM, based on a vast cohort of patients from a 20-year time period. We are showing for the first time that SGA is found in different predilected anatomic areas than SVMs, and their clinical and imaging characteristics are significantly different. Low-flow SVMs present in imaging as cystic lesions, while SGA always presents in a homogenous epifascial cap shape. Recognizing these imaging characteristics should enable clinicians to diagnose SGA without requiring further invasive diagnostic workup.

## Figures and Tables

**Figure 1 children-10-00362-f001:**
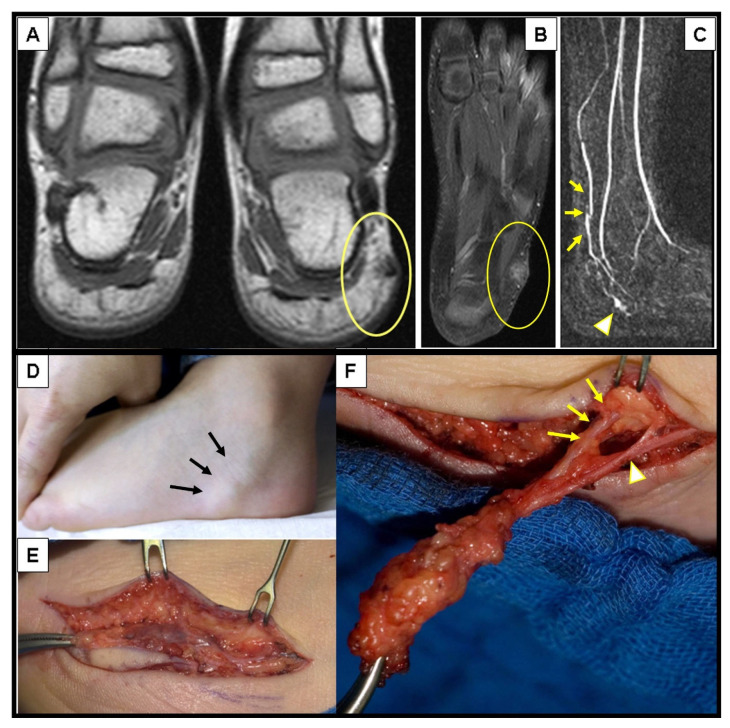
A 3-year-old boy with a pulsatile lesion under the malleolus lateralis of the left foot (**D**)—(black arrows). MRI showed a localized arterio-venous malformation in the subcutaneous tissue (**A**,**B**) with an AV shunt in the MR-angiography (**C**)—(arrowhead) depicted by the early visualization of the vein (**C**)—(yellow arrows). Complete surgical excision was performed (**E**) with ligation of the feeding artery (**F**)—(yellow arrows) and the draining veins (**F**)—(arrowhead).

**Figure 2 children-10-00362-f002:**
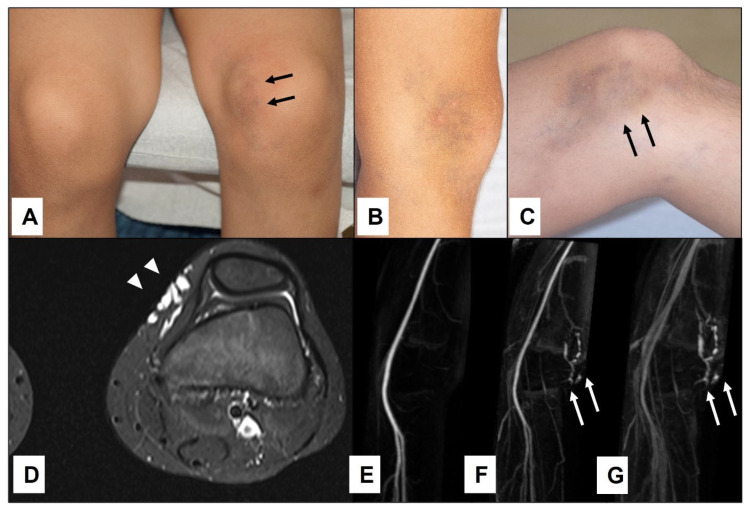
An 11-year-old boy presented with bluish swelling medially on the left knee (**A**): white (arrowheads). The lesion was compressible and would drain and flatten with the elevation of the leg indicating clinically the venous malformation (**B**,**C**)—(white arrowheads). MRI showed the exact location and extension of tubular-cystic venous malformation (**D**)—(white arrowheads), which filled with contrast material in the venous phase of the MR angiography (**E**–**G**)—(white arrows), confirming the clinical diagnosis. The patient was referred to sclerotherapy.

**Figure 3 children-10-00362-f003:**
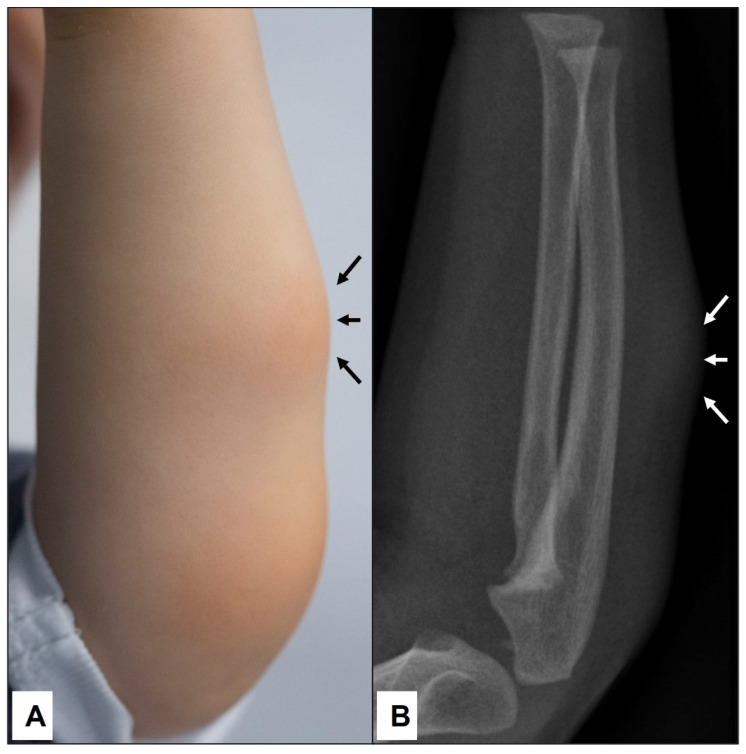
A 2-year-old boy was presented by his parents due to the swelling on the left forearm, which persisted for the last 2 months (**A**)—(black arrows). Because the subcutaneous lump was firm and immobile, an X-ray was conducted to exclude bone abnormalities (**B**). Note the homogenous soft tissue expansion in the subcutaneous area (white arrows) but no osseous involvement.

**Figure 4 children-10-00362-f004:**
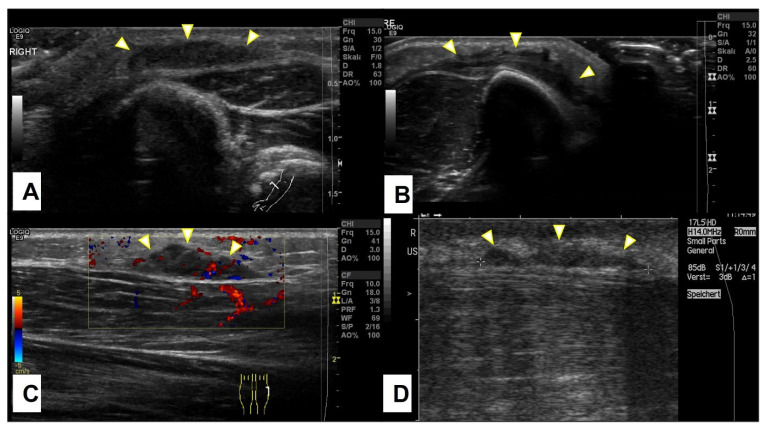
Representative slices of ultrasound imaging of subcutaneous granuloma annulare (SGA). The lesions are strictly epifascial with hypo- and hyper-echogenic zones and mild hypervascularization. A common finding is the cap shape of these lesions, marked with arrowheads. The depicted lesions were located on (**A**) the right forearm; (**B**) the right lower leg; (**C**) the left lower leg; (**D**) the right lower leg.

**Figure 5 children-10-00362-f005:**
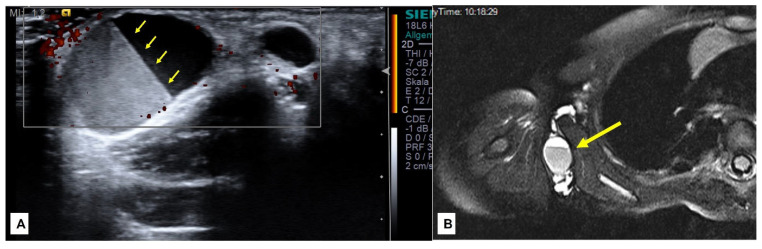
A 2.5-year-old girl with a sudden appearance of a lump in the right axilla. Ultrasound examination showed a multi-cystic mass with intracystic bleeding documented by the fluid-fluid level in the cyst representing the clot retraction (**A**)—(yellow arrows). The MRI (**B**) showed the exact extension of the mass (arrow), which was limited to the subcutaneous tissue and facilitated the indication for surgical resection.

**Figure 6 children-10-00362-f006:**
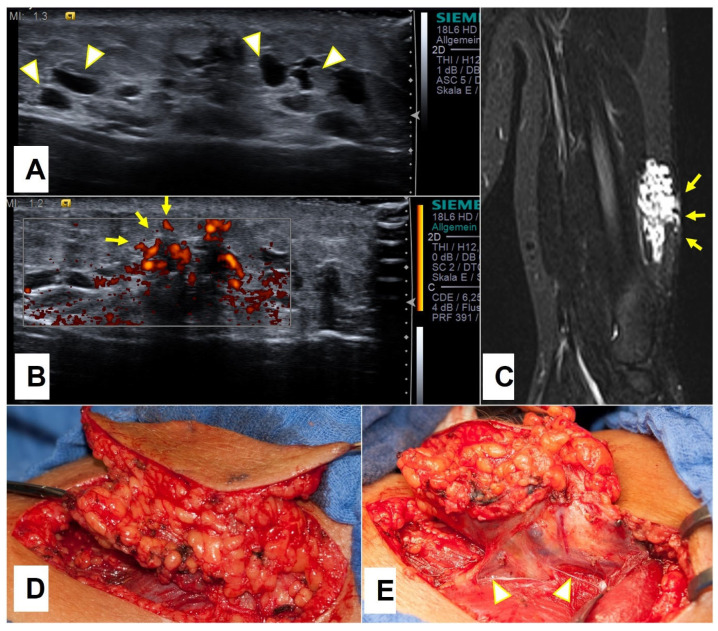
A 14.5-year-old boy was presented because of the minor swelling on his left upper arm, which appeared after he started to go to the gym. Ultrasound imaging showed a tubulo-cystic lesion (arrowheads) on the lateral aspect of his upper arm (**A**) which showed perfusion (arrows) in a power Doppler mode with venous flow characteristics (**B**). MRI depicted a venous malformation (arrows) of the localized extension located in the subcutaneous area (**C**). Complete excision of the VM was performed (**D**). Draining veins are depicted with arrowheads (**E**).

**Figure 7 children-10-00362-f007:**
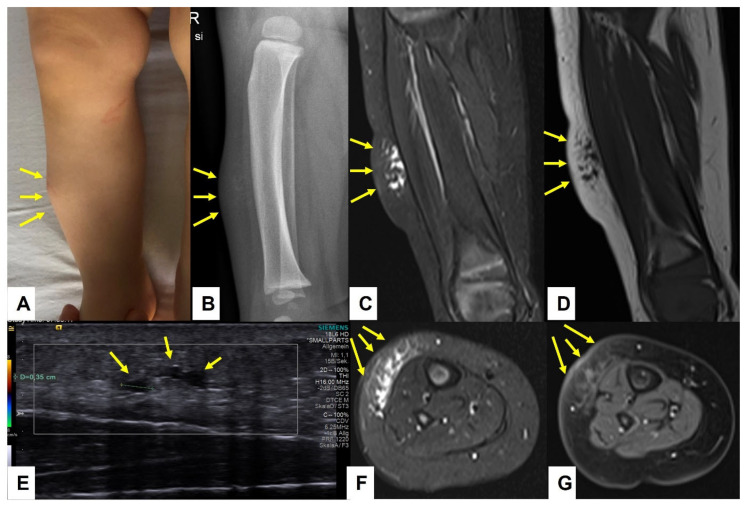
This composite figure illustrates a subcutaneous lump on the right lower leg of a 12-month-old girl (**A**)—(arrows). X-ray at presentation showed a cloudy pretibial soft-tissue lesion with no osseous deformities (**B**)—(arrows). Ultrasound imaging of the lesion was unspecific but showed some small cystic areas in the deep dermis (**E**)—(arrows). MR imaging at the age of 13 months showed a strict epifascial tubulo-cystic lesion in the subcutaneous tissue in the anterolateral area without contrast enhancement (**C**,**D**,**F**,**G**)—(arrows).

**Figure 8 children-10-00362-f008:**
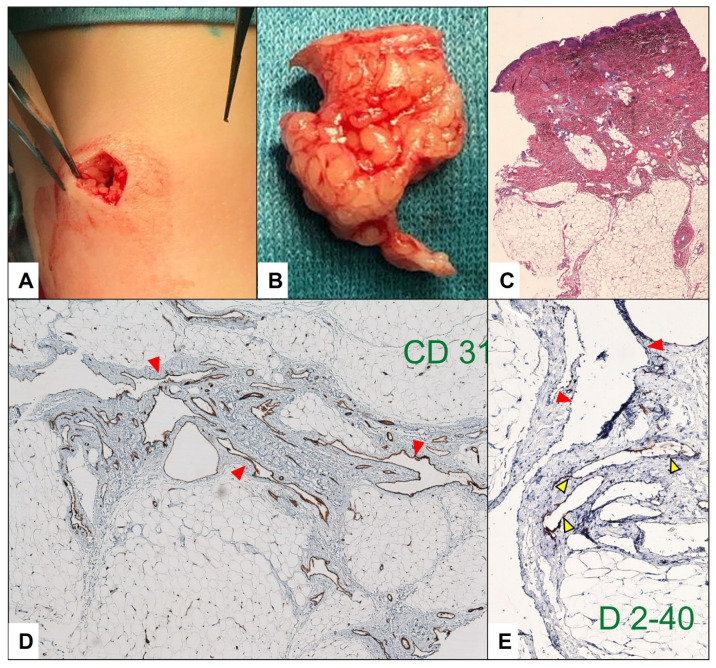
The patient from Figure 7 underwent an incisional biopsy of the pretibial lesion at 15 months of age (**A**,**B**). The H&E staining showed no granulomatous tissue in specimen (**C**). Immunohistochemistry showed irregular, dilated vascular structures surrounded by fat tissue in the deep subcutaneous tissue layer, which strongly expressed the CD31 vascular marker (**D**)—(red arrowheads). D2-40 (**E**) stains positive in the lymphatic component (yellow arrowheads) and is negative in the endothelial cells of the veins (red arrowheads).

**Figure 9 children-10-00362-f009:**
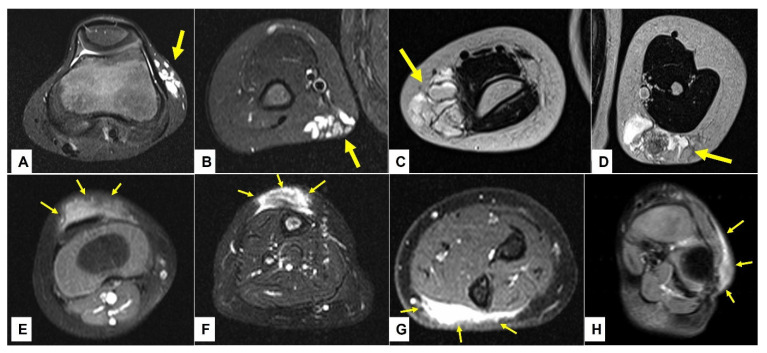
This composite figure illustrates the typical imaging differences between the low-flow subcutaneous vascular malformations (SVM) and subcutaneous granuloma annulare (SGA) as seen on the MR images. In images (**A**–**D**), four patients with multicystic subcutaneous lesions pointed out by yellow arrows are presented. These lesions were accurately diagnosed by MRI and after excision, also confirmed by histopathology as a venous malformation on the left knee in an 11-year-old boy (**A**); venous malformation on the right upper arm in a 4-year-old boy (**B**); lymphatic malformation on the right elbow in a 4-year-old girl (**C**); lymphatic malformation on the left upper arm in an 8-year-old girl (**D**). In images (**E**–**H**), four patients with homogenous subcutaneous lesions marked with 3 yellow arrows each are presented. These lesions remained inconclusive after MRI, with the main differential diagnosis being the low-flow SVM in all cases. Because malignancy could not be excluded, surgical biopsy was needed, and the diagnosis of SGA was confirmed by histopathology in all cases. (**E**) right knee in a 2.5-year-old boy, (**F**) right lower leg in a 3-year-old girl, (**G**) left forearm in a 4-year-old boy, (**H**) left foot in a 3-year-old girl. Note that all SVM have a cystic appearance on the MRI, whereas SGA shows the typical epifascial extension with gradual rise of the lesion towards the more superficial tissues, which we have named the “epifascial cap” sign. These self-limiting lesions show a homogenous appearance in the MRI and do not invade the fascia.

**Figure 10 children-10-00362-f010:**
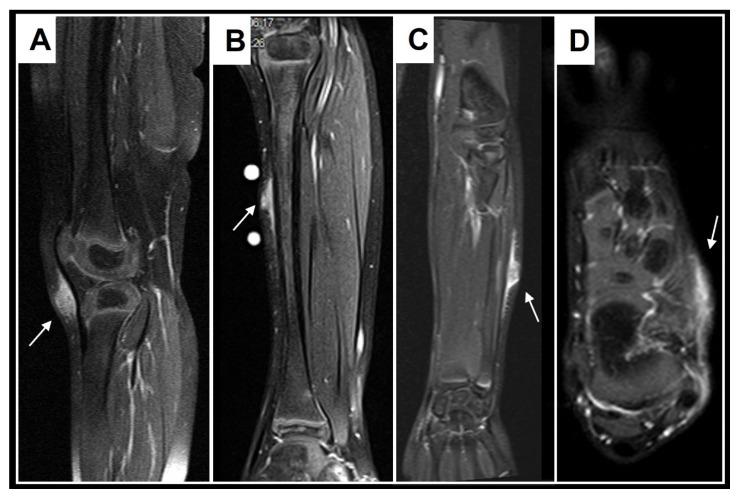
These images show representative MRI slices through SGA (indicated by white arrows) at the knee (**A**); lower leg (**B**); forearm (**C**); and foot (**D**) in four different cases. Note that all SGA lesions show the typical epifascial extension with gradual rise of the lesion towards the more superficial tissues, which we have named the “epifascial cap” sign.

**Figure 11 children-10-00362-f011:**
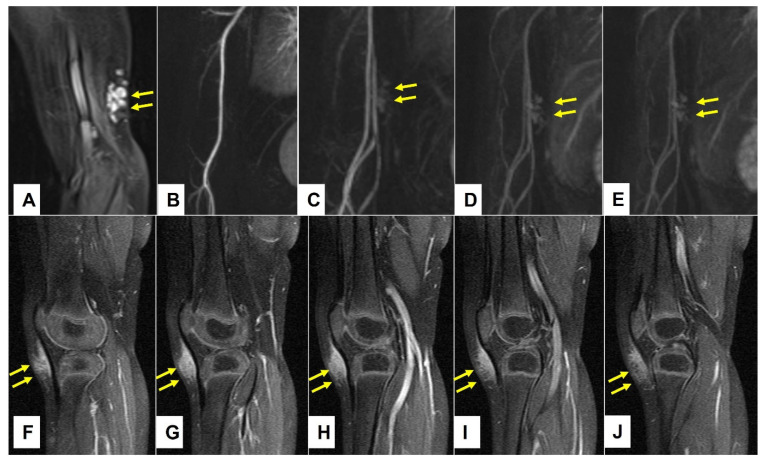
This composite figure illustrates the typical differences in contrast enhancement between the subcutaneous venous malformations (VM) and subcutaneous granuloma annulare (SGA) during MRI. The lesions are pointed out by yellow arrows. Image (**A**) depicts the extension of the VM on the medial side of the right upper arm. The lesion cannot be seen during the arterial phase of MR angiography (**B**). The VM fills slowly at the beginning of the venous phase at 32 s post-contrast application (**C**) and intensifies at 96 (**D**) and 120 s (**E**) post-contrast application. The SGA on the anterior side of the right knee, as depicted in image (**F**), shows a homogenous enhancement after contrast application which slowly fades away at the end of the examination (**G**–**J**).

**Figure 12 children-10-00362-f012:**
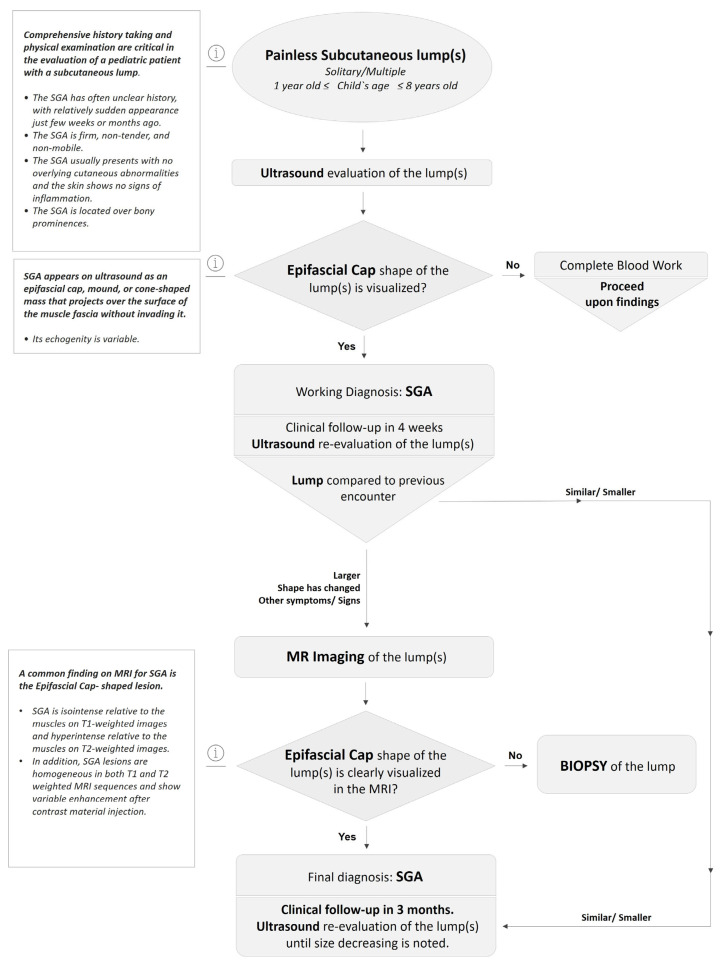
Management algorithm for Subcutaneous Granuloma Annulare in children.

**Table 1 children-10-00362-t001:** Characteristics of patients with subcutaneous granuloma annulare (SGA) and low-flow subcutaneous vascular malformations (SVM) who underwent MR imaging.

	N	Age in Years—Median (Range)	Girls N (%)	Surgical Intervention N (%)
**SGA**	12	3.25 (2–5)	9 (75%)	12 (100%)
**SVM**	47	5 (0–18)	19 (40%)	45 (96%)

Legend: SGA = subcutaneous granuloma annulare; SVM = subcutaneous vascular malformations; N = number of patients.

**Table 2 children-10-00362-t002:** Location of the lesions in patients with subcutaneous granuloma annulare (SGA) and low-flow subcutaneous vascular malformations (SVM) who underwent MR imaging.

	Total N (%)	Head and Neck N (%)	Upper Extremity N (%)	Lower Extremity N (%)	Trunk N (%)
**SGA**	12 (100%)	3 (25%)	3 (25%)	6 (50%)	0
**SVM**	47 (100%)	6 (13%)	23 (51%)	8 (19%)	10 (17%)
LM	27 (57%)	1	16	3	7
LVM	8 (17%)	3	3	1	1
VM	12 (26%)	2	4	4	2

Legend: SGA = subcutaneous granuloma annulare; SVM = subcutaneous vascular malformations; LM = lymphatic malformation; LVM = lymphatico-venous malformation; VM = venous malformation; N = number of patients.

## Data Availability

The raw data supporting the conclusions of this article will be made available by the authors without undue reservation.

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
