# Peer review of "Subcutaneous Granuloma Annulare vs. Subcutaneous Vascular Malformations in Children: A Diagnostic Challenge"

_children, 2023, doi:10.3390/children10020362_

Round 1
Reviewer 1 Report
This is an interesting study and I do think that the authors' take home message regarding the utility of the "epifascial cap" to suggest a diagnosis of SGA. However, there is too much additional detail and far too many figures showing clinical examples of SVMs which are irrelevant. I would recommend editing this manuscript to hone in on the aforementioned primary finding.
Would recommend referring to SGA simply as "SGA" rather than "SGA lumps." Would also avoid the use of nonspecific terms such as "lesion" where possible.
Author Response
This is an interesting study and I do think that the authors take home message regarding the utility of the "epifascial cap" to suggest a diagnosis of SGA. However, there is too much additional detail and far too many figures showing clinical examples of SVMs which are irrelevant. I would recommend editing this manuscript to hone in on the aforementioned primary finding.
Dear reviewer, we appreciate your comments and opinion. As the topic of this
manuscript has not been the focus of the scientific literature so far, we have decided to illuminate it, in order to stimulate other researchers to pay more attention to these fascinating clinical issues. Not only SGA but also vascular malformations located in subcutaneous layer of the body (SVM) have not been studied well in the literature. There are no data about their prevalence among the entity of vascular malformations, and no clear algorithms about their management. With examples presented in the figures we tried to explain the procedures in cases with SVM which were excluded from the analysis, and also show examples showing clinical and imaging differences in the diagnostic process between patients with SGA and patients with SVM. We hope that our study may stimulate other researchers to analyze their experience with this diseases.
Would recommend referring to SGA simply as "SGA" rather than "SGA lumps."Would also avoid the use of nonspecific terms such as "lesion" where possible.
Thank you very much for this important comment, which we followed during the
complete manuscript, and which indeed helps omit unnecessary words.
The word lump was deleted 12 times.
The word lesion(s) was deleted 36 times.
Reviewer 2 Report
Comments to the Authors of Manuscript Number children-2168591
with full title
Subcutaneous granuloma annulare versus subcutaneous vascular malformations in children: a diagnostic challenge
This is an excellent manuscript that presented a complex analysis of clinical and imaging diagnosis of subcutaneous granuloma annulare and subcutaneous vascular malformations. Small additions are necessary.
1. Line 81-82: please explain in the text the abbreviations ISSVA, AVM, VM, LM.
2. At line 381 there is an editing error: ? is the error.
Recommendation: Minor Revision.
Author Response
This is an excellent manuscript that presented a complex analysis of clinical and
imaging diagnosis of subcutaneous granuloma annulare and subcutaneous vascular malformations. Small additions are necessary.
Dear reviewer, we appreciate your kind judgement.
1. Line 81-82: please explain in the text the abbreviations ISSVA, AVM, VM, LM.
Abbreviations explained, thank you!
2. At line 381 there is an editing error: ? is the error.
Editing error corrected, thank you!